# Assessing Supply Chain Performance from the Perspective of Pakistan's Manufacturing Industry Through Social Sustainability

**Maryam Khokhar [1], Wasim Iqbal [2], Yumei Hou [1,*], Majed Abbas [1] and Arooj Fatima [1]**

1 College of Economics and Management, Yanshan University, Qinhuangdao 066004, China; maryamkhokhar@stumail.ysu.edu.cn (M.K.); abbasmajed@gmail.com (M.A.); Aroojfatima132@yahoo.com (A.F.)

2 Department of Management Science, College of Management, Shenzhen University, Shenzhen 518060, China; wasimiqbal01@yahoo.com

* Correspondence: hym@ysu.edu.cn

**Abstract:** The industry is gradually forced to integrate socially sustainable development practices and cross-social issues. Although researchers and practitioners emphasize environmental and economic sustainability in supply chain management (SCM). This is unfortunate because not only social sustainable development plays an important role in promoting other sustainable development programs, but social injustice at one level in the supply chain may also cause significant losses to companies throughout the chain. This article aimed to consolidate the literature on the responsibilities of suppliers, manufacturers, and customers and to adopt sustainable supply chain management (SSSCM) practices in the Pakistani industry to identify all possible aspects of sustainable social development in the supply chain by investigating the relationship between survey variables and structure. This work went beyond the limits of regulations and showed the status of maintaining sustainable social issues. Based on semi-structured interviews, a comprehensive questionnaire was developed. The data was collected through a survey of the head of the supply chain in Karachi, Pakistan. The results of this paper showed that organizational learning was the most important dimension of supplier social sustainability with a value of 40.5% as compared to the effectiveness of the supply chain and the supplier performance with values 37.7 and 9.6%, respectively. In terms of the manufacturer's social responsibility, the highest score for operational performance was 47%, while productivity was 20%, and corporate social demonstration was 20%. Finally, for the customers' social sustainability, two dimensions were determined, namely, customer satisfaction and customer commitment with scores of 47 and 40%, respectively. We also solved sustainable social problems from the perspective of suppliers, manufacturers, and customers. The study would help professionals anywhere to emphasize their considerations and would improve the management of social sustainability in their supply chain.

**Keywords:** sustainable supply chain management (SSSCM); social sustainability; qualitative research; Pakistan

## 1. Introduction

Due to the rigidity of the industrial environment, communities, Non-Government Organizations (NGOs), and consumer awareness policies are under pressure. Organizations must implement sustainable implementation in supply chain management. Sustainable development combines economic, social, and environmental characteristics and transcends limits within and between industries. Therefore, sustainable business is directly related to the sustainable supply chain management (SSSCM)

proposition [1]. Sustainability is defined by contextualization [2], "the ability to meet the needs of the next generation without compromising today's needs". Besides, SSSCM can be defined as "how to manage social issues that can sustain long-term strategies of the organization".

The above aspects are not limited to the internal operations of suppliers, manufacturers, and customers, but also can be extended to external social issues of the organization. The environmental and economic sustainability has a considerable impact on the literature and practice. However, social sustainability has not received enough attention. In order to enhance sustainability and other aspects of practice, a socially sustainable industrial scale needs to be accurately achieved. Most researchers focus on how the industry develops social sustainability when working with upstream or downstream companies [3]. Despite much research on South Asian countries, there is less evidence of Pakistani industries with different social norms. Some advocates emerging and protection of labor rights and how these measures support claims for greater efficiency [4]. To date, few studies have incorporated social factors into their SSSCM framework and have not incorporated sustainable management practices. Several cases have attempted to address the sustainability of the business sector with a short-term focus on social initiatives undertaken by the organization. These attempts are not conducive to improving social measures and building the capabilities and resources needed to comprehensively and systematically manage the social impact of the organization's supply chain management. Some studies have taken the first step in identifying and examining some useful issues and aspects related to social sustainability.

The purpose of this study was to investigate social sustainability in Pakistan from the perspective of major companies, first-tier suppliers, and first-line customers. This research made a significant contribution to the existing literature. First, we extended the current literature by testing and validating models to enhance Pakistan's sustainable supply chain management practices through social commerce motivations, mechanisms, and performance results. Second, our data analysis revealed significant differences between regulatory constraints that indicated a position to maintain sustainable social issues. Third, this research raised awareness that in the context of social commerce, socially sustainable management in its supply chain is still crucial. Fourth, our research provided empirical evidence that standard adaptation practices of manufacturing companies would positively affect customers' willingness to buy in social enterprises.

## 2. Theoretical Background

### 2.1. Sustainable Supply Chain Management (SSSCM)

Sustainable supply chain management can often be described as the processes and practices that are carried out within and across organizations to achieve emotional synchronization to increase output, reduce costs, maximize asset utilization, and maximize customer service. The transactions involve its activities, resources, information funds, and the impact of the supply chain on the social well-being of its employees, society, and customers. It also minimizes environmental impacts [5]. Increasingly, companies are held accountable for the social, economic, and ecological decisions that arise from their internal and supplier operations [6]. For the past two decades, SSSCM has worked to integrate social, economic, and environmental goals through focused business processes. It has become a practice for companies to achieve sustainable development results in their supply chains [7]. However, the management register is not able to inspire global SSSCM. SSSCM is recommended to increase stakeholder attention to the impact of social enterprise internal supply chain operations on social systems [8].

The business sector performance can be improved by the sustainability of the supply chain. This directly affects the competitiveness of the industry and the performance of the supply chain. Koberg and Longoni [9] adopted a plan to solve social problems at multiple levels. According to Wan Ahmad et al. [10] in order to minimize waste and save costs, the plan must be implemented with sustainable motivation, and suppliers and manufacturers should be strengthened through the

refinement between the capabilities and assistance of internal representatives. In order to implement SSSCM, the social, economic, and environmental requirements of industry models and practices have been proposed. According to Yuen et al. [11], in order to improve the sustainability of the business sector, its long-term goal is to manage the well-being of ordinary people who must control social and economic management operations. This is why many industries use sustainability indices to assess their social sustainability capabilities [12].

### 2.2. Social Sustainable Supply Chain Management (SSSCM)

With regard to the environmental and economic sustainability of industrial practices, SSSCM should also be considered when companies are seeking successful and sustainable growth. Social sustainability should be managed through social issues to improve the long-term life of the industry [13]. These issues are important characteristics of sustainable corporate development that are being considered and evaluated [14,15]. Former scholars use the same report to address social issues in the supply chain. Determining the scope and measures of universal and global social sustainability is challenging because of the lack of conceptual clarification, especially in South Asian countries like Pakistan, where it involves social issues. It is, therefore, clear that supply chain managers do not have a sufficient understanding of the social issues involved and how to evaluate and manage [16].

A review of the existing literature shows that there are huge challenges in addressing SSSCM and social issues related to the business sector; few studies have explicitly and broadly focused on the dimensions of social issues and sustainability [17]. According to research by Moroke et al. [18] further research is necessary to observe the social sustainability dimensions of Pakistan. Therefore, this article focused on the social sustainability of supply chain management of the business sector. Figure 1 illustrates the conceptual model of the study.

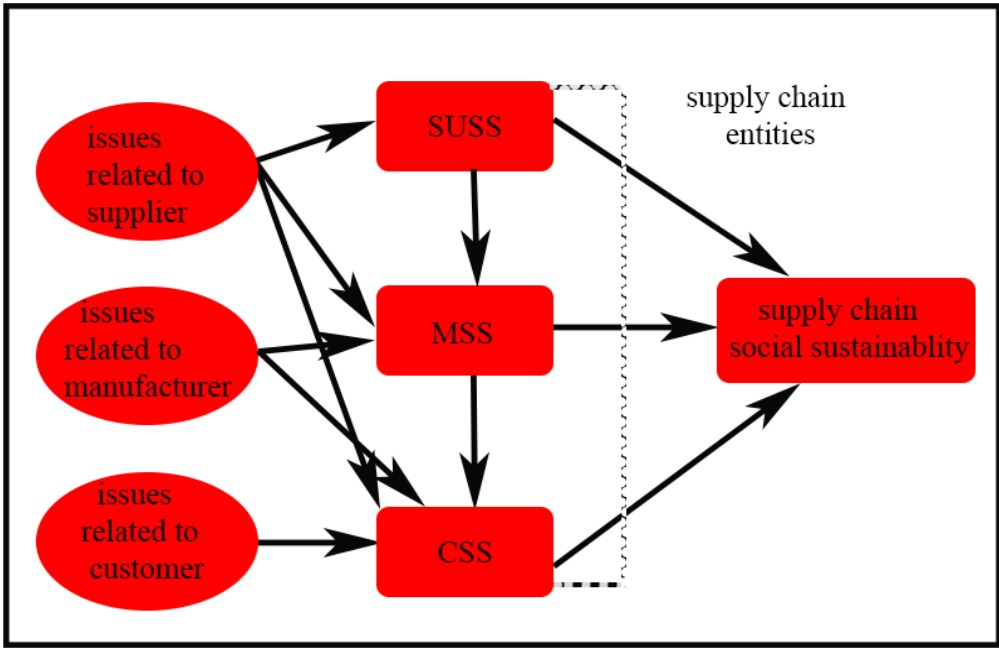

**Figure 1.** Conceptual model of this study. Note: SUSS: supplier social sustainability, MSS: manufacturer social sustainability, and CSS: customer social sustainability.

### 2.3. Social Sustainable Supply Chain Issues and Dimensions

Socially sustainable supply chain practices include managing social issues in all three stages of the supply chain. Social issues fall into two categories: basic issues (good for health and safety) and progress issues (reporting goods and practices) [19]. Ollila and Macy [20] suggested that improved organizational social issues should include simple basic questions: What do you say about social

issues? Whom are you fighting for? Is it a mechanism? Our previous arguments refute the previous query needs for basic and progressive social issues. The response to the next query comes from the reorganization of complex individuals in the supply chain. Stakeholder perceptions interpret customers, manufacturers, suppliers, governments, and society as stakeholders in the industry and the right of a company's activities to affect their health and safety benefits. As such, stakeholders play a vital role in leveraging the company's social and sustainable management practices [21]. Analysis changes social issues at various stages of the supply chain and stakeholders [22]. For example, supplier social sustainability issues and dimensions, manufacturer social sustainability issues and dimensions, and customer social sustainability issues and dimensions are very powerful. Besides, these dimensions are directly related to society.

### 2.3.1. Supplier Social Sustainability

Sustainable supply chain management (SSCM)audits usually carefully consider suppliers, social issues, and dimensions, which always marks the complexity of individuals in supply chain management [23]. Parsa, Roper, Muller-Camen, and Szigetvari [24] used case studies to find improved labor rights and their impact on the supply chain from a supplier's perspective. In contrast, Sendlhofer and Lernborg [25] classified labor rights based on the perception of suppliers in South Asian countries. Others have demonstrated the organization's commitment to management, occupational safety, health management, wages, education and training, labor relations, worker welfare, research and development, ethics, and children and slavery and their role in sustainable supply chains [26]. Taking into account the above issues of suppliers, it can help to achieve the social sustainability of upstream supply chains [27].

Others believe that the implementation of social sustainability means many influential services in the upstream supply chain and their relationships with suppliers, jobs, and customer representatives. Merad et al. [28] validated the positive link between middle management and customer pressure and the sustainable development of society. However, their research did not establish a positive correlation between management pressure and the implementation of social sustainability. Similarly, Fortunati and O'Sullivan [29] found that imitation, regulation, and daunting pressures exist in the implementation of sustainable social development. Once a supporter of the principles of sustainable development, it is deeply entrenched that companies implement socially sustainable development policies. Further the supply chain social issues identified through different researcher are mentioned in Table 1.

**Table 1.** List of supply chain social issues identified through literature.

| Social Issues | Suppliers | Manufacturer | Customer | Literature |
|---|---|---|---|---|
| Child labor and forced labor | X | X | X | [30,31] |
| Diversity | X | X | X | [32,33] |
| Discrimination | X | X | X | [34,35] |
| Health and Safety | X | | X | [36,37] |
| Unethical practice | X | X | X | [38,39] |
| Philanthropy | X | X | X | [40,41] |
| Labor practices | X | X | | [39,42] |
| Human rights | X | X | | [43,44] |
| Wages | X | X | | [45,46] |
| Education | X | X | X | [46] |
| Sustainable sourcing | X | X | | [32,47] |
| Local sourcing | X | X | | [48,49] |
| Product responsibility | X | X | X | [50,51] |
| Employee welfare | X | X | X | [32] |
| Employment creation | | X | | [20] |
| Poverty alleviation | | X | | [29] |
| Local economic development | | X | | [32,33] |
| Stakeholders engagement | | X | | [50,51] |

### 2.3.2. Manufacturer's Social Sustainability

In addition, considering social issues at the manufacturing level strengthens the overall sustainability of key businesses. Manufacturers' social sustainability requires the management of social issues that theoretically plague individuals, workers, society, and customers. These issues are complex in manufacturing practice. Through the case study process, based on stakeholder awareness, Ikram et al. [52] identified organizations' management, occupational safety and health management, wages, labor rights, education and training, labor relations, children and slavery, organizational commitment, worker welfare, research and development, altruism, stakeholders, product liability, and social issues in the Pakistani Industry and their role in the SSSCM. It is also believed that true statements of sustainable development practices for stakeholders and consumers tend to take a gradual approach and expand consideration of the business sector [53,54].

The value of social sustainability promotes manufacturing management's responsibility to provide equal opportunities, positive scope, and refinement of natural life, which is spread throughout the community. Some opinions favor the development of appropriate education and training resources, fair policies, and worker welfare. Quality management in manufacturing can better manage social issues that deserve improvement. MARTIN [55] discussed 14 points on quality development, work environment, self-improvement plan, fearless job training, and fair income, which are essential to increase production value. Other practical studies have established a link between quality improvement, employee satisfaction, knowledge improvement, and inclusion programs. The organization's commitment to management, occupational safety and health management, and wages and labor rights is positively related to the strategic presentation of companies, which demonstrates what is observed among stakeholder's status and reputation. Subsequently, based on speculative support provided by stakeholders, Mani et al. [56] discussed child labor and forced labor practices, diversity, altruism, employee safety, welfare, and ethical issues and how to focus on solving these problems in your business. The convenience and other performance of social representatives are considered welfare.

### 2.3.3. Customer's Social Sustainability

Customer social sustainability points to social issues in the downstream indicators of the supply association, focusing primarily on the personal issues of customers and suppliers. Generally, it falls into two categories. The former contract is concluded by social issues that prevail at the sellers, which mark the health and safety benefits of the working class, which, in turn, affects trade administration and forms a concern for the entire supply chain [57]. The issue of the next contract to use manufactured items establishes the foundation for end-consumer health and safety issues and empowers the company's sustainable implementation. Among the shared actual customers, downstream associations play a leading role in their dominant position as they stimulate manufacturers and suppliers.

In previous work, customer-related social sustainability issues are broadly related to health and safety management, social issues, customer rights, and education [58]. These issues are gradually affecting executive representatives [59]. In a study of Pakistan (Atef et al.) [60] vendors considered various social issues, including health management, social issues, customer rights, education, corruption and bribery, job creation, ethical labels, and respect for customer privacy that affects SSSCM. Through previous work, the characteristics of social sustainability point to diversity, ethical issues, health and safety, human rights, and job creation. However, the semi-characteristics and events of companies are different, and their attitudes to these issues in business activities are also different [61]. Regardless of the increasing literature on suppliers, sustainable social development plays a role, but this information system is quietly developing and becoming a prerequisite for further consideration in emerging economies [62].

This theoretical background recognizes the need to study social sustainability standards in South Asian countries, especially Pakistani industries, from the perspectives of key companies, front-line traders and consumers. The different Dimensions of Social Sustainability in Supply Chain according to the different researcher point of view are presented in Table 2.

**Table 2.** Dimensions of social sustainability in the supply chain according to literature (scale items and measures for social sustainability) [63].

| Measures | Items Measures | Source |
|---|---|---|
| Organizational commitment to management (suppliers and manufacturers) | OCM1 | [64,65] |
| Occupational safety and health management (suppliers, manufacturers, and customers) | OCH1 | [66] |
| Wages (suppliers, manufacturers, and customers) | WS1 | [67] |
| Labor rights (suppliers and manufacturers) | LR1 | [63] |
| Customer issues (customers) | CI1 | [64] |
| Educational training (customers) | ET1 | [68] |
| Labor relations (suppliers and manufacturers) | IR1 | [69] |
| Stakeholder (manufacturers) | ST1 | [70] |
| Child and bonded (suppliers and manufacturers) | CB1 | [71] |
| Worker welfares (suppliers and manufacturers) | WW1 | [32] |
| Research and development (suppliers and manufacturers) | RD1 | [72] |

This theoretical background recognizes the need to study social sustainability standards in South Asian countries, especially Pakistani industries, from the perspectives of key companies, front-line traders, and consumers.

## 3. Methodology

### 3.1. Measurement Development

In this study, we tested 41 projects on a pilot scale with 19 supply chain manufacturing managers. In developing and completing the questionnaire, we followed the generally accepted recommendations on wording issues [73]. Appendix A lists the measurement items and their related sources.

We used a five-point Likert scale (1 totally disagrees, 5 completely agrees). The Likert scale has been used in several sustainability measurement studies [74]. To ensure the content is valid, we conducted expert reviews to improve the tool. All construction projects were originally developed in English.

It is a Ph.D. student who is proficient in English who participated in the translation process. The original Urdu questionnaire was tried by some of our colleagues and online friends. Before being accepted as the final version, 40 useful responses were returned. Our model included several control variables to ensure that empirical results were not biased due to covariance between variables.

### 3.2. Survey Design

We designed a survey to test our research hypotheses and conduct it on the Pakistani industry. We chose the survey method because this quantitative research predicts behavior and examines the relationship between variables and construction. In line with cheap labor and the government's pursuit of creating an encouraging environment for manufacturers, Pakistan's sustainability assessment ranks six of the most ideal industrial goals. Therefore, the Pakistan Business Council in 2005 recognized various factors that promote industrial effectiveness and sustainability. To collect survey data, a pilot test was conducted for this study.

Our target group included 19 supply chain manufacturing managers and experts in facial expression effectiveness and readability, including general manager, assistant general manager, senior manager, CEO, and VP (sustainability), who participated in the bi-annual supply chain management IIMB Conference held in December 2018. In addition, the Pakistani Corporate and Regulatory Affairs and the Pakistan Security and Transaction Commission (SECP) have ordered fully registered organizations to observe and issue a Business Responsibility Report (BRR) horizontally through its business statement. Figure 2 shows the industries targeted in our survey and the percentage of participating companies in each industry. Appendix A lists the industries and all the related companies.

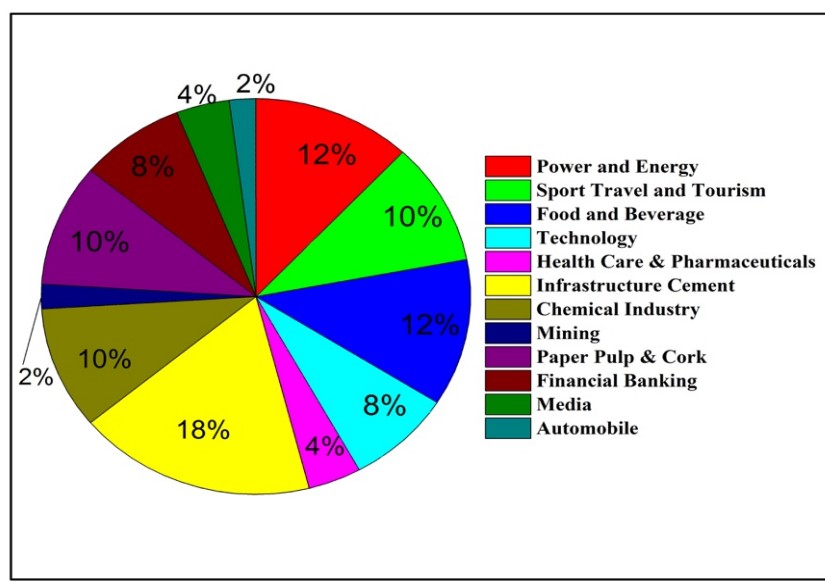

**Figure 2.** The share of industries targeted in our survey.

### 3.3. Data Collection

We conducted semi-structured interviews to accumulate data. Closed-ended questioning and semi-structured interviews helped to reach core rationality by confirming that a comparative assessment of the responses of all respondents was made. A pre-test was conducted to assess the authenticity of the interview etiquette, and then a pilot test was conducted with the supply supervisor. The semi-structured form was sent to 12 different industries, and its release address can be found on its official website link.

We invited an administrative representative to confirm that the questionnaire was filled by supply chain operations, procurement, logistics, and production supervisors. In the sample structure, all respondents who received pre-tests belonged to different industries (as described below) and were not sponsors of the main research. The corresponding supply chain executives representing various industries in Pakistan participated in preliminary tests to complete the semi-structured questionnaire. Managers were selected based on their experience; they should have at least five years of experience in the supply chain and sustainable development of society. The data collected was triangulated with supplementary data sources (i.e., print media, company reports, retired members of the company) to define the extent to which respondents in the trial structure were able to truly answer the questions asked. After making minor changes to the interview etiquette, we determined that the protocol (Appendix A) and trial structure were appropriate for this study.

The trial structure consisted of Pakistan's well-known supply chain directors. We sought to contribute to the most familiar and respected supervisors. Therefore, our trial structure involved special envoys or spokespersons who have been requested at the Pakistan Supply Chain Association (SCAP) meetings over the past few years. Respondents' data were collected by the conference leader. The 25 directors were contacted primarily based on the technical level, industry, and corporate income. Overall, 11 members answered preliminary appeals. The interview was scheduled in May 2019. The honorary member carried archived data and other credentials on sustainable social efforts in his supply chain.

### 3.4. Data Analysis

Data analysis was done by having each respondent mark their opinions within a given dimension. Each questionnaire was based on respondents' industry experience. To improve response times, an alternate email was sent (twice a week). Eventually, 503 responses were collected from 12 industries by ordinary post. From the 503, 412 (81.90%) were selected for post-evaluation, and 91 (18.10%) were

rejected due to incomplete data or information. After each conversation, a comprehensive summary was organized, recording key facts identified by each participant. The applicant's true identity was clarified by phone. The interview was then translated; the meeting and archive information were carefully observed to discover the subject. To improve internal coherence, an interview with an expert in supplier sustainability education was conducted individually. Interviewers and other experts were recorded separately and interviewed. Analyzing all conversations could lead to trivial deviations. To improve external effectiveness, this approach was generalized to the research setting [75]. This research used contributors from various fields of the Pakistani industry. In addition, we introduced the members by translating interview information and conclusions in order to obtain a response on the representativeness and authenticity of the data.

## 4. Results and Discussion

### 4.1. Dimensions of Supplier Social Sustainability Analysis

We used the induction-deduction method in our analysis. Our analysis yielded a variety of topics related to the adoption of suppliers, focal companies, and customers. If these practices were cited in the report, they were cited as implementations for that particular company. First, various social sustainability topics were identified through a literature review. Based on the literature review, a social sustainability taxonomy for the supply chain was established. This helped us identify the keywords used to run the query. By comparing the sustainable supply chain management with socially sustainable supply chain management practices mentioned in the literature review and the report, we identified patterns and themes related to the practices adopted by Pakistani companies. At the same time, word frequency and potential coding techniques led to the identification of new topics that were not part of the listed taxonomy.

Job satisfaction describes various aspects of supplier social sustainability and related issues. Organizational commitment also has been broadly covered in management research. The members highlighted activities, such as those related to increasing personal productivity, increasing effort, and increasing job satisfaction, which are important features of supplier SSSCM. Other characteristics, such as reduced absenteeism and employment of workers in the organization, were grouped together and labeled under "organizational commitment to management" (OCM). The results in Table 3 indicated that the dimensions related to OCM were 2.4% high individual productivity, 35.3% increased endeavor, 28.7% higher job satisfaction, 13.5% less absenteeism, and 20.1% worker employment in an organization being the most significant. Asrar-ul-Haq et al. [76] stated the importance of job satisfaction and organizational commitment to supply chain sustainability. The issue of corporate responsibility for management is even more common in Pakistan's supply chain. Existing expansions are further straining supply-side supply chain management (tier 2, 3, etc.), as smaller companies tend to use organization-based ideals and rules to reduce absenteeism and worker employment.

Participants also discussed issues related to occupational health and safety management conditions, occupational hazard prevention, and the implementation of 28% safety and health roles (called "health and safety") in supplier workplaces. The safety and security of 28% of female employees were essential for the reporting of cumulative incidents via social media. Social issues related to security were also highlighted in 19.9% of the total responses related to occupational safety and health management (OSHM). Finally, issues related to ensuring compliance with regulations and the measurement of health and hygiene status were highlighted in 20.9% of the responses.

Some contributors have defined the importance of labor rights (13.3%) [77]. Forced labor and human trafficking (35.8%) are recorded in Pakistan. Another important dimension is child labor, i.e., children's engagement in hazardous work and debt bondage. Many executives have suggested banning child labor and restricted labor. One manager explained that the minimum age for employment was observed as 47.9%, child abuse was prohibited, and old-age benefits were paid recorded as 41.9%.

**Table 3.** Supplier social sustainable supply chain management dimensions.

| Constructs | Items | Description | F | P | C |
|---|---|---|---|---|---|
| Organizational commitment to management | OCM1 | High individual productivity | 10 | 2.4 | 2.4 |
| | OCM2 | Increased endeavor | 149 | 35.3 | 37.7 |
| | OCM3 | Higher job satisfaction | 121 | 28.7 | 66.4 |
| | OCM4 | Less absenteeism | 57 | 13.5 | 79.9 |
| | OCM5 | Worker employment in an organization | 85 | 20.1 | 100 |
| Occupational Safety and health management | OSH1 | Occupational hazards prevention and implementation of safety and health roles | 118 | 28 | 28.6 |
| | OSH2 | Maintaining the safety measurement for women at workplace | 122 | 28.9 | 58.3 |
| | OSH3 | Ensuring that the company complies with the regulations | 84 | 19.9 | 78.6 |
| | OSH4 | Measurement of health and hygiene conditions | 88 | 20.9 | 100 |
| Wages | W1 | Reasonable wages paid to employees | 157 | 37.2 | 38.1 |
| | W2 | No wages period should exceed one month | 199 | 47.2 | 86.4 |
| | W3 | Not violates labor laws | 56 | 13.3 | 100 |
| Labor rights | LR1 | Forced labor and human trafficking is illegal | 151 | 35.8 | 36.7 |
| | LR2 | No child under the age of fourteen years engaged in any hazardous work | 230 | 54.5 | 92.5 |
| | LR3 | Secure the well-being of people | 31 | 3.7 | 100 |
| Educational training | ET1 | Encourage employees to become productive | 174 | 41.2 | 42.2 |
| | ET2 | Reduce workplace injuries and accidents | 207 | 49.1 | 92.5 |
| | ET3 | Produce a sense of responsibility | 31 | 7.3 | 100 |
| Industrial relations | IR1 | Vast access to the whole seller | 112 | 26.5 | 27.2 |
| | IR2 | Local and international exposure | 145 | 34.4 | 62.4 |
| | IR3 | Right of the collective bargaining | 62 | 14.7 | 77.4 |
| | IR4 | Right of the strike or power to go slow | 93 | 22.2 | 100 |
| Child and bonded | CB1 | Follow the rule with the minimum age for employment | 202 | 47.9 | 49 |
| | CB2 | Child abusing is prohibited | 210 | 49.8 | 100 |
| Worker welfare | WW1 | Provision for the old-age benefits grant | 177 | 41.9 | 43 |
| | WW2 | Social security for workers | 156 | 37 | 80.8 |
| | WW3 | Compensation in case of sickness, maternity, injury, or death | 79 | 18.7 | 100 |
| Research and development | RD1 | Adopt R&D culture in future growth prospect | 210 | 49.8 | 51 |
| | RD2 | Valid through products and process innovation | 202 | 47.9 | 100 |

Participants also described how to use "sweet shops or violate two or more labor laws". In practice, the labor force in small cities is more disintegrating, in violation of two or more labor laws, sweatshop labor, reasonable wages paid to employees, lower typical substandard working conditions, and wages provided. Managers stressed the wage period must not exceed one month, with minimal compensation to maintain workers and sustainability. The supply chain manager agreed.

Supply chain managers often considered the role of education and skills improvement. These pieces of training included health and safety, hygiene, achievement, and capacity development in new careers. Scholar Conaty and Robbins [78] emphasized the impact of reducing payroll injuries and accidents, creating a sense of responsibility, extensive contact with the entire seller, and workers' educational benefits on supplier performance and supply chain management practices. However, in South Asian countries, funds are still included in education accounts because it means that traders and suppliers can save more money. Here, suppliers are mainly required to invest in employers' health and safety hygiene practices training.

Applicants emphasized on worker welfare to improve social sustainability. Although issues, such as workers' social security and sickness, maternity leave, compensation for injuries or deaths, appear to be similar in South Asian countries and especially in Pakistan, suppliers in these countries differ in engaging in such activities. The suppliers have adopted a culture of R&D, which is anticipated to be 49.8% in future growth prospects. They have also effectively modernized the temple through product and process innovation, and the impact of donated schools and hospitals on elementary schools on social vendor representatives corresponds to 47.9%. Despite the analysis of humanitarian charities, a study has confirmed different philanthropic measures in South Asian countries.

## 4.2. Dimensions of Manufacturer Social Sustainability

Metrics related to manufacturers' social sustainability were not sufficient for an explicit assessment of the primary business and transient environment. Table 4 provides a list of the manufacturers' social dimensions of sustainability derived from the given data. Workers emphasized on social activities, such as adding local suppliers, buying from local suppliers, buying women-owned plans, supporting different communities in building hospitals, schools, and colleges, skilled training centers, conducting worker training, and seeking employment. Others discussed the importance of building health centers, hospitals, and health camps to improve social health. The importance of building public canters to promote social well-being and expand support for sustainable social agriculture was also discussed as a means to improve social and community sustainability. In addition, managers discussed the importance of establishing portable drinking water facilities in cities because too many workers cannot drink pure water. Workers with higher effective commitment were calculated to be 67.8%, which was very high, as shown in Table 4.

**Table 4.** Manufacturer social sustainable supply chain management dimensions.

| Constructs | Items | Description | F | P | C |
|---|---|---|---|---|---|
| Organizational commitment to management | OCM1 | Workers with higher effective commitment | 286 | 67.8 | 70.1 |
| | OCM2 | Normative commitment involves a feeling of the moral obligation of workers | 126 | 29.9 | 100 |
| Occupational safety and health management | OSH1 | Integrated affected mechanisms designed | 160 | 37.9 | 38.8 |
| | OSH2 | Control the risk that may affect worker's health and safety | 168 | 39.8 | 79.6 |
| | OSH3 | Define certain policies for women's safety at the workplace | 84 | 19.9 | 100 |
| Wages | W1 | Reasonable wages paying to employees | 328 | 77.7 | 79.6 |
| | W2 | Wages are to be paid within seven days after the end of the wage period | 84 | 19.9 | 100 |
| Labor rights | LR1 | Ensuring appropriate labor working conditions | 253 | 60 | 61.4 |
| | LR2 | Work rights are exceptional from cast, creed, or race | 159 | 37.7 | 100 |
| Educational training | ET1 | For skill enhancement | 127 | 30.1 | 30.8 |
| | ET2 | For development | 126 | 29.9 | 61.4 |
| | ET3 | For accelerating productivity | 117 | 27.7 | 89.8 |
| | ET4 | For high supervision | 42 | 10 | 100 |
| Industrial relations | IR1 | There are critical factors in industrial relations unitary, pluralist, Marxist, and radical | 160 | 37.9 | 38.8 |
| | IR2 | Based on a healthy relationship between employee and employer | 252 | 59.7 | 100 |
| Child and bonded | CB1 | Child under the age of 15 is prohibited at workplace | 244 | 57.8 | 59.2 |
| | CB2 | Employers are incurred not by the children themselves | 168 | 39.8 | 100 |
| Worker welfare | WW1 | Housing (employer-provided or employer-paid) | 85 | 20.1 | 20.6 |
| | WW2 | Furnished or not | 84 | 19.9 | 41 |
| | WW3 | With or without free utilities | 84 | 19.9 | 61.4 |
| | WW4 | Group insurance (health, dental, life, etc.) | 126 | 29.9 | 92 |
| | WW5 | Disability income protection | 33 | 7.8 | 100 |
| Research and development | RD1 | R&D regards the effort increases the pressure to perform in case of failure | 286 | 67.8 | 69.4 |
| | RD2 | Plan developed under R&D more productive than the current plan | 126 | 29.9 | 100 |
| Altruism | A1 | Increases another person's welfare, belief that the others are equally treated | 143 | 33.9 | 34.7 |
| | A2 | Traditional virtue | 126 | 29.9 | 65.3 |
| | A3 | Communities and supplier to behave ethically | 143 | 33.9 | 100 |
| Stakeholders | S1 | Demand for sustainable measures to the society | 96 | 22.7 | 23.3 |
| | S2 | Seek to understand strategically integrated issues | 144 | 34.1 | 58.3 |
| | S3 | Impact the firm sustainability | 124 | 29.4 | 88.3 |
| | S4 | Emerging economies | 48 | 11.4 | 100 |

Another activity included disseminating employment opportunities for qualified young people to activists in response to activists' past complaints that major companies did not serve 29.9% of young people. Issues, such as health and hunger, adequate housing, and employment formation,

have been explored in developing countries. For developing countries, other issues, such as helping society with sustainable agriculture, drinking water facilities, and establishing primary healthcare centers, are unique characteristics. More discussions about workers are "teaching for the professional development of workers" or "training for the efficiency of industrial institutions", and the main focus is on workers' sustainability education.

A large number of participants (37.9%) drew attention to health and safety procedures. The rest 39.8% advocated the ethical role of manufacturers in protecting contract workers. However, they were not entitled to permanent positions. In addition, some people mentioned the cleanliness of the organization, hoping to provide a healthy atmosphere for employees. For the benefit of cooperation, 19.9% of executives emphasized on women's health and safety issues. Our research found social issues related to the right to health and security in developing countries, especially Pakistan.

Supply chain managers emphasized ethical aspects that did not allow employees to participate, such as environmental pollution, coercion, bribery. Gender discrimination in employment, transfer, and promotion was also highlighted. This research provided organizations with a platform on which they can develop social sustainability by applying human methods, such as donating to cancer hospitals, religious organizations, NGOs, orphanages (sweet homes), schools refurbishment, and cultural heritage donations.

During the interview, many executives highlighted issues related to child labor and labor, as shown in Table 4. The social participation of children in labor activities is sad, which is very high (57.7%) in Pakistan. Finally, the issues that are most reflected at the enterprise level are often ensuring sanitation in the surrounding area, youth unemployment, the construction of primary healthcare centers, health awareness seminars, and drinking water facilities in public places. Companies may investigate to ignore local needs to identify major issues.

*4.3. Dimensions of Customer Social Sustainability*

Table 5 shows a list of dimensions related to the social sustainability of customers. Unsurprisingly, many customer-related issues were comparable to social issues related to suppliers and manufacturers. From the perspective of SSSCM, customers had mainly contributed to business-to-business (B2B) customers. However, some interviewees could plan carefully when dealing with end consumers. Based on our research data, preventing child-parent relationships and protecting human rights suggested key aspects of research. In developing countries, such as Pakistan, both issues support the entire system.

Respondents emphasized the importance of using non-toxic resources that could harm customers' health, and these were organized by H & S and recorded as just 37.5%. Respondents judged social issues by declaring responsive packaging, using appropriate product labels, non-toxic resources used in packaging, and maintaining customer H & S during product use recorded as 13%. They also discussed the social issues of building customer objection and response tools. Managers worried that healthcare protection for channel workers must be linked to supply chain management representatives. Gender diversity in the appointment and promotion of channel staff was also highlighted. Supply chain managers emphasized and discussed training to adapt workers to capacity growth and business development. Several managers' training of employees was full of attention to employee maintenance and sustainability.

**Table 5.** Social Sustainable dimensions of customers.

| Constructs | Items | Description | F | P | C |
|---|---|---|---|---|---|
| Safety and health management | SHM1 | The place must be hazards free for visiting customers | 179 | 37.5 | 43.4 |
| | SHM2 | Risk prosecution | 62 | 13.0 | 58.5 |
| | SHM3 | May lose customer tendency | 171 | 35.8 | 100 |
| | SHM4 | Access to fresh drinking water, corruption, native violence, drug use | 46 | 9.6 | 11.2 |
| Social issue | SI1 | Environmental contamination | 180 | 37.7 | 54.9 |
| | SI2 | Inadequate emergency services | 87 | 18.2 | 76.0 |
| | SI3 | Inequality, poverty, racism | 99 | 20.8 | 100 |
| Customer rights | CR1 | The right information about the product | 169 | 35.4 | 41 |
| | CR2 | Right to have accessibility for considering substitutes | 90 | 18.9 | 62.9 |
| | CR3 | Defense from false and distorted rights in advertising labeling and observing | 153 | 32.1 | 100 |
| Customer issues | CI1 | Social sustainable and supply chain management practices impact cooperation in their strategic and operational performance | 170 | 35.6 | 41.3 |
| | CS2 | Customer's social sustainability is positively associated with local businesses' supply chain performance | 242 | 50.7 | 100 |
| Education | E1 | Increase awareness | 196 | 41.1 | 47.6 |
| | E2 | Emphasized for product moves | 127 | 26.6 | 78.4 |
| | E3 | Awareness of product issues that cause safety and health issues | 89 | 18.7 | 100 |

### 4.4. Discussion and Implications

Table 6 contains a description of each dimension and its relative frequency. These frequencies indicated the level of socially sustainable entities and their importance to corporate productivity. Mangers insisted that there was a proportional relationship between supplier performance and time management of goods (delivery time) with less turbulence. This would lead to a healthy and worry-free environment. An executive said that as social sustainability is adopted, all of the above factors would reduce work stress, minimize operational risks, and improve product quality and the ability to meet buyer needs.

**Table 6.** Dimension wise results and processes (supplier, manufacturer, and customer).

| Construct | Items | Results and Related Process | F | P | C |
|---|---|---|---|---|---|
| Supplier social sustainability | SSS1 | Organizational learning: improved collaboration between suppliers, manufacturers, and customers | 190 | 40.5 | 46.1 |
| | SSS2 | Effective supply chain: manufacture value improvement and timely delivery to buyers | 177 | 37.7 | 89.1 |
| | SSS3 | Supplier performance: capable against buyers, fewer fluctuations in supply | 45 | 9.6 | 100.0 |
| Manufacturer social sustainability | MSS1 | Productivity: raised yield per worker, the adaptation of new technology | 94 | 20.0 | 22.8 |
| | MSS2 | Corporate social demonstration: ethical business execution, productivity, trustworthy suppliers | 96 | 20.5 | 46.1 |
| | MSS3 | Operational performance: competence, quality products, and consistency | 222 | 47.3 | 100.0 |
| Customer social sustainability | CSS1 | Customer association and commitment: enhance attentiveness, improved communication skill | 188 | 40.1 | 45.6 |
| | CSS2 | Customer performance: enlightened patience, increased sale, time management skill development | 224 | 47.8 | 100.0 |

The social sustainability of the organization helped to leverage operational performance, quality, and reliability, thereby enhancing the company's social representation and promoting results. An executive added the company's social description. Considering all shareholders' research findings on customer social issues, the company's image was enhanced. New customer acquisition and loyalty related to social sustainability practices. This was a continuous process. The executive concluded that the method originally used in the study was to conduct a preliminary inspection of SSSCM and explore

key businesses, front-line traders, and consumers by applying the social sustainability dimension. Secondly, social problems were attributed to the consequences of socially sustainable development and analyzed in terms of dimensions. Next, the aspects and results of social sustainability in different industries in Pakistan were investigated.

Compared to environmental and economic sustainability, our research provided the social characteristics of sustainability, which is not practiced at all [79]. Promoting discussions about social sustainability is a daunting task because it benefits the well-being and personal prestige of society. Even in terms of sustainability, emphasis should be placed on, for example, "organizational commitment to customer management, occupational safety and health management, customer comfort, wages, labor rights for ethical production, education with product accessibility, industrial relations, children and slavery, worker welfare, research, development, and contribution to society".

Considering the social sustainability of suppliers, the results were linked to [79], which establishes a bond and plays a role in the formal recognition and firm conviction between supplier sustainability measures mediating role. Nonetheless, researchers from [78] only focused on the realization of suppliers and purchasing functions, the process of social sustainability, and this research proposed a key business of social sustainability, the front-line trader, and consumer side. In addition, by applying ethical values, the ethical behavior exhibited by suppliers also recognizes the purpose of achieving company sustainability). We emphasized that moral values were related to the social sustainability of developing countries. This finding was in line with the results and the research direction of [78], but we improved their work by suggesting the implementation of social sustainability standards in developing countries. We, therefore, agreed to explore more discoveries about the social sustainability of developing countries.

Considering the social sustainability of manufacturers, most activities are mainly related to the business of the organization and the social responsibility of stakeholders and industry representatives. These findings lay the foundation for cooperation between social and environmental sustainability for the future and recognize the diversity and characteristics of SSSCM integration in developing countries and measures and help export and commercialize social representatives.

This study proposed numerous activities related to joint results and the social sustainability of clients. Our findings were related to [77], who was accustomed to building a link between corporate reputation and the sustainable performance of customer society. This research has brought new insights into the phenomenon of social sustainability and recommended a more comprehensive study of sustainable supply chain management, including suppliers, manufacturers, and customers.

## 5. Conclusions and Limitations

This study discussed many dimensions of SSSCM in Pakistan's manufacturing industries. These dimensions addressed the social issues in SSSCM and were distinctly different from advanced economies. The study raised many social questions about how companies can retain social assets, theoretically improve sustainability, and differentiate on a participant basis. In addition, the study revealed consequences in social expressions, such as sustainable implementation, how it mimics trade procedures. This study gave serious attention to the SSSCM literature by providing insights into various social issues and their scale, results, and experiments in developing countries. The ensuing social issues and scale of sustainable development were related to the industrial social supply chain and provided decision-making services to supply chain managers, who proposed to establish socially accessible supply chains in developing countries. In addition, the results and processes of socially sustainable supply chain management were also presented.

Nonetheless, the study had some limitations. We used data collected from several trade industries in Pakistan. Although the number of pilots was not too large, the demographics of the applicants (in terms of participation in the senior management industry, number of participants, and percentage of participants) were not very different from the source. This approach needs to improve overall social issues by selecting applicants in certain industries, and it must also be believed that, over time,

treating positions diagonally must provide contributors with different altitudes and different groups of skill. However, further research should follow the example settings. We assume that Pakistan has representatives of several emerging countries.

However, conducting this research can test or develop our findings through assessments in other developing countries. Analysis can also explore the links between trade and social sustainability outcomes. In addition, improving the sympathetic relationship between environmental and social dimensions is an important area of research, but it has not responded much to past research. For future consideration and positive quantitative analysis, these findings can be used to examine the strength of the proposed multidimensional social sustainability model. For this reason, it is reasonable to conduct a preliminary study on the effect or importance of each standard size. Finally, results and processes related to the social sustainability dimension are reported. Advanced studies can be added to confirm these and their recommended associations.

**Author Contributions:** Conceptualization, M.K. and Y.H.; Methodology, M.K.; Software, W.I.; Validation, M.A. and A.F.; Investigation, W.I.; Resources, Y.H.; Data Curation, Y.H.; Writing-Original Draft Preparation, M.K.; Writing-Review & Editing, M.A., M.K.; Visualization, A.F.; Supervision, Y.H.; Project Administration, Y.H.; Funding Acquisition, Y.H. All authors have read and agreed to the published version of the manuscript.

**Funding:** We highly praise the honorable Professor Hou Yumei for her valuable guidance and great support in this research. This research was supported and funded by the project Joint Optimization of Omni-Channel Retailer Procurement and Pricing Considering Consumer Behavior (G2019203387) under the umbrella of Hebei Province Natural Science Foundation Project in 2019.

**Acknowledgments:** The author would like to thank the five anonymous reviewers for their helpful comments. This has greatly improved the performance of the paper.

**Conflicts of Interest:** The authors declare that there is no conflict of interest regarding the publication of this paper.

**Interview Decorum:** We thank all the interviewees for their valuable opinions (valuable opinions on the supply of goods (raw materials), manufacturing process, and customer behavior). We hope to conclude that there is no right or wrong statement. We have observed positive and negative responses and found useful feedback. We are trying to analyze the social sustainability processes in your organization's supply chain.

## Appendix A

**Table A1.** Industries demographics and companies.

| Power and Energy | Code | Sports, Travel, and Tourism | Code | Food and Beverages | Code | Technology | Code |
|---|---|---|---|---|---|---|---|
| PAEC (Pakistan Atomic Energy Commission) | A1 | PKSF (Pakistan Kettlebell Sports Federation) | B1 | Dalda Foods Pvt Ltd | C1 | Suparco | D1 |
| PEM (Pakistan Energy Mix) | A2 | UNWTO (Developing Tourism Industry of Pakistan) | B2 | Mair Foods | C2 | AI and IoT for Pakistan | D2 |
| Ministry of Energy | A3 | PTDC (Pakistan Tourism Development Corporation) | B3 | Mitchell's Fruit Farms Limited | C3 | Reditus | D3 |
| EIA: Electricity | A4 | Kashmir Pakistan Tourism | B4 | Murree Brewery | C4 | TECH Pakistan | D4 |
| AEMC (Atomic Energy Medical Center) | A5 | TAAP | B5 | National Food | C5 | | |
| CREA (Center for Research on Energy and Clean Air) | A6 | | | OMORE | C6 | | |

| Healthcare and Pharmaceuticals | Code | Infrastructure and Cement | Code | Chemical Industries | Code | Mining | Code |
|---|---|---|---|---|---|---|---|
| PPMA | E1 | WEP | F1 | Solvay | G1 | Pakistan International Bulk Terminal Limited | H1 |
| SEARLE | E2 | Refhankaisha Hankuri Foundation | F2 | CUF | G2 | | |
| | | PCF | F3 | ICL | G3 | | |
| | | Jeevan Welfare Foundation | F4 | ICI Pakistan | G4 | | |
| | | Lucky Cement | F5 | Descon | G5 | | |
| | | Askari Cement Ltd. | F6 | | | | |
| | | Bestway Cement Ltd. | F7 | | | | |
| | | Attock Cement Pak Ltd. | F8 | | | | |
| | | Kohat Cement Company Ltd. | F9 | | | | |

| Paper, Pulp, and Cork | Code | Financial, Banking, and Insurance | Code | Media | Code | Automobile | Code |
|---|---|---|---|---|---|---|---|
| FARAH INTERNATIONAL, Karachi | I1 | NBP | J1 | Dawn News | K1 | Suzaki | L1 |
| FARUKI PULP MILL LTD, Lahore | I2 | MCB | J2 | Geo News | K2 | | |
| ASIA CELL INTERNATIONAL, Karachi | I3 | Bank Alfalah | J3 | | | | |
| IPO Pakistan | I4 | Meezan Bank | J4 | | | | |
| PFVA | I5 | | | | | | |

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
