# Peer review of "Assessing Supply Chain Performance from the Perspective of Pakistan’s Manufacturing Industry Through Social Sustainability"

_processes, doi:10.3390/pr8091064_

Round 1

Reviewer 1 Report

The paper assesses sustainable supply chain performance in the Pakistani manufacturing sector through a literature review and interview. The paper requires extensive changes to be considered for publication.

  1. Kindly proofread the paper with the help of a native English speaker. Grammar mistakes can be seen throughout the paper. Consider changing the title too.
  2. The abstract doesn't show the results of the study, kindly include the major findings.
  3. Table 1 in-text citation, Carter and Jennings - year is missing.
  4. The methodology section is totally ambiguous. What is the application of PLS SEM in this study? Where are the results of PLS SEM? 
  5. Figure 2 is not legible.
  6. The conclusion should highlight the main findings of the study.
  7. The structure of the paper makes it difficult for readers to follow. Kindly consider changing or improving it before resubmission.

Good luck!!

Author Response

Reviewer #1

Comments

Responses

1. Kindly proofread the paper with the help of a native English speaker. Grammar mistakes can be seen throughout the paper. Consider changing the title too.

Thank you for your time and comments. We appreciate your comments. The paper has been improved in terms of English language. We have refined the title as per your invaluable comments.

2. The abstract doesn't show the results of the study, kindly include the major findings.

Thank you for your suggestions. We have include the major findings in the abstract as per your suggestions.

3. Table 1 in-text citation, Carter and Jennings - year is missing.

Thank you for your suggestions. We are very sorry for our mistake. We have enlisted the year at missing place.

4. The methodology section is totally ambiguous. What is the application of PLS SEM in this study? Where are the results of PLS SEM?

Thank you for your time and comments. The application of PLS SEM in this study is used in result and discussion part to find out the percentage, frequency and cumulative of table 4, 5, 6 and 7.

5. Figure 2 is not legible.

Thank you for your suggestions to improve the paper. We have addressed all the comments given in the attach file. We revised the manuscript and deleted Figure 2, because there is no need to do so.

6. The conclusion should highlight the main findings of the study.

Thank you for your suggestions to improve the paper. Discussion has been improved. The conclusion has been highlight the main findings of the study and revised as per your invaluable suggestions.

7. The structure of the paper makes it difficult for readers to follow. Kindly consider changing or improving it before resubmission.

Thank you for your suggestions to improve the paper. We have addressed all the comments given in the attach file and the paper has been revised thoroughly.

Reviewer 2 Report

The topic of this study is interesting. However, it needs a huge improvement to better highlight the main contribution.

  • English language and style should be highly revised.
  • The theoretical background is very relevant for the manuscript development. However, at now, it is quite redundant (e.g., lines 100-104 and 118-124) and comments remains too general (e.g., at line 130, which are the challenges in addressing SSCM?). It seems that the same concepts are repeated several times with very few additional details. Moreover, it includes also some literature review aspects. I suggest to revise this section by condensing the text making clearer the key point without repetition throughout the different subsections and to provide more details on the relevant aspects (e.g., challenges, benefits, barriers, ...) and if appropriate to provide a separate section for the literature review. At line 245, you stated the 'literature review'. I think that a shorter and clearer section on the theoretical background and a more detailed section on the literature review can improve the readability of the manuscript.
  • Define all the acronyms (e.g., at line 38, SCSS).
  • Table 1 check the style of the caption. I also suggest to provide the references as rows and the social issues as column to avoid repetition. Table 1 is in the supplier subsection, why in the table all the supply chain actor are included?
  • All the table are not cited in the text. It is not clear why Table 2 is put in line 215. Provide a cross-reference and check where the table are placed.
  • The methodology should be better explained in a structured way, and supported by references.
  • Provide graphs for defining the sample (e.g., pie graphs showing the industries and the role of the interviewed). This increases the provided information and allows to reduce the text.
  • Which kind of questions have been used in the interviews? give some insights on them.
  • Figure 2 is not clear. Check style of the caption
  • Section 4.1, the results of the literature review have been used to develop the questionnaire or as separate results? It is not clear whether the results are deduced form the literature review or from the interviews.
  • References that you might find interesting: doi:10.3390/en10101618, doi:10.1016/j.indmarman.2013.10.002, doi:10.1016/j.ijpe.2010.11.010, doi:10.1108/SCM-12-2013- 0440

Author Response

Reviewer #2

Comments

Responses

• English language and style should be highly revised.

Thank you for your suggestions to improve the paper. We appreciate your comments. The paper has been improved in terms of English language.

•The theoretical background is very relevant for the manuscript development. However, at now, it is quite redundant (e.g., lines 100-104 and 118-124) and comments remains too general (e.g., at line 130, which are the challenges in addressing SSCM?). It seems that the same concepts are repeated several times with very few additional details. Moreover, it includes also some literature review aspects. I suggest to revise this section by condensing the text making clearer the key point without repetition throughout the different subsections and to provide more details on the relevant aspects (e.g., challenges, benefits, barriers, ...) and if appropriate to provide a separate section for the literature review. At line 245, you stated the 'literature review'. I think that a shorter and clearer section on the theoretical background and a more detailed section on the literature review can improve the readability of the manuscript.

Thank you for your suggestions to improve the paper. The paper has been revised thoroughly. We have addressed all the comments given in the attach file. The sentence in Line100-104 and 118-124 have been revised in the revised manuscripts. Line 130, are also revised. We are very sorry for our mistake. Also, thank you for your time and comments. We have corrected the concepts which are repeated several times. We have done all corrections in the revised manuscripts. Line No. 245, need has been addressed. We have added the data regarding theoretical background and a more detailed section on the literature review in the revised manuscripts. 

•Define all the acronyms (e.g., at line 38, SCSS).

Thank you for your suggestions to improve the paper. We revised the sentence in line 38 of the revised version of SCSS, which is SSCM (Sustainable Supply Chain Management), and all comments are given in the attachment.

•Table 1 check the style of the caption. I also suggest to provide the references as rows and the social issues as column to avoid repetition. Table 1 is in the supplier subsection, why in the table all the supply chain actor are included?

Thank you for your suggestions to improve the paper. We have addressed all the comments (especially about table 1) given in the attach file and the paper has been revised thoroughly. Table 1 is in the supplier subsection, in the table all the supply chain actor are not included just these are included, employment creation, Poverty alleviation, Local economic development and Stakeholders engagement.

• All the table are not cited in the text. It is not clear why Table 2 is put in line 215. Provide a cross-reference and check where the table are placed.

Thank you for your suggestions to improve the paper. We appreciate your comments. All manuscript with the table citation has been improved and revised as per your invaluable suggestions.

•The methodology should be better explained in a structured way, and supported by references.

Thank you for your time and comments. Various researchers have done methodological and application studies by this way.

•Provide graphs for defining the sample (e.g., pie graphs showing the industries and the role of the interviewed). This increases the provided information and allows to reduce the text.

Thank you for your time and comments. Various researchers have also defined the same type of research without any pie graphs. All results will also be discussed in the table and text to show the respondents' industry and role.

•Which kind of questions have been used in the interviews? Give some insights on them.

Five-point Likert scale (1 totally disagrees, 5 completely agrees) open handed kind of questionnaires we have used in the interviews. The Likert scale also has been used in several sustainability measurement studies. To ensure the content is valid, we conduct expert reviews to improve the tool. All construction projects were originally developed in English.

•Figure 2 is not clear. Check style of the caption

  Thank you for your suggestions to improve the paper. We deleted Figure 2 about your valuable comments. And view all the graphics and table styles in the revised manuscripts.

•Section 4.1, the results of the literature review have been used to develop the questionnaire or as separate results? It is not clear whether the results are deduced form the literature review or from the interviews.

Thank you for your suggestions to improve the paper. In Section 4.1, the results are from Table 4, and these results were obtained through interviews.

•References that you might find interesting: doi:10.3390/en10101618, doi:10.1016/j.indmarman.2013.10.002, doi:10.1016/j.ijpe.2010.11.010, doi:10.1108/SCM-12-2013- 0440

Thank you for your suggestions to improve the paper. We get help from these papers and we have added the references that are related to the comments given in the attach file.

Round 2

Reviewer 1 Report

The authors have edited the paper, however I dont feel that a considerable change has been made to be qualified to publish in the journal.

There are typos in the newly added parts as well - Organizational learning is the most important dimension of supplier social sustainability recognized by 40.5% of the contributors compared to 37.7 % motioned effectiveness of supply chain and 9.6 mentioned supplier performance.

Figure 1 - What is SUSS, MSS, CSS? 

Please go through the paper again for grammar corrections and minor typos.

Author Response

Reviewer #1

Comments

Response

There are typos in the newly added parts as well - Organizational learning is the most important dimension of supplier social sustainability recognized by 40.5% of the contributors compared to 37.7 % motioned effectiveness of supply chain and 9.6 mentioned supplier

Thank you so much sir for your time and for the improvement of paper.

Sir we try to overcome the errors that you suggested.

The results of this paper show that organizational learning is the most important dimension of supplier social sustainability with value 40.5% as compared to the effectiveness of supply chain and the supplier performance 37.7% and 9.6% respectively. In terms of manufacturer’s social responsibility, the highest score for operational performance is 47%, while productivity is 20% and corporate social demonstration is 20%. Finally, for the customers social sustainability, two dimensions were determined namely customer satisfaction and customer commitment with scores of 47% and 40% respectively.

Figure 1 - What is SUSS, MSS, CSS? 

Thank you so much sir for improvement of manuscript

SUSS: supplier social sustainability, MSS: manufacturer social sustainability and CSS: customer social sustainability

Please go through the paper again for grammar corrections and minor typos.

Sir we have double check the grammatical errors and try to remove minor typos

Reviewer 2 Report

Previous comments have been addressed by the Authors.

Author Response

Reviewer #2

Thank you very much sir, with your suggestions and help, we improve the manuscript.